# Gorilla: Large Language Model Connected with Massive APIs

**Shishir G. Patil**[1][*]    **Tianjun Zhang**[1][*]    **Xin Wang**[2]    **Joseph E. Gonzalez**[1]

[1]**UC Berkeley**    [2]**Microsoft Research**

shishirpatil@berkeley.edu

## Abstract

Large Language Models (LLMs) have seen an impressive wave of advances, with models now excelling in a variety of tasks, such as mathematical reasoning and program synthesis. However, their potential to effectively use tools via API calls remains unfulfilled. This is a challenging task even for today's state-of-the-art LLMs such as GPT-4 largely due to their unawareness of what APIs are available and how to use them in a frequently updated tool set. We develop Gorilla, a finetuned LLaMA model that surpasses the performance of GPT-4 on writing API calls. Trained with the novel Retriever Aware Training (RAT), when combined with a document retriever, Gorilla demonstrates a strong capability to adapt to test-time document changes, allowing flexible user updates or version changes. It also substantially mitigates the issue of hallucination, commonly encountered when prompting LLMs directly. To evaluate the model's ability, we introduce APIBench, a comprehensive dataset consisting of HuggingFace, TorchHub, and TensorHub APIs. The successful integration of the retrieval system with Gorilla demonstrates the potential for LLMs to use tools more accurately, keep up with frequently updated documentation, and consequently increase the reliability and applicability of their outputs. Gorilla's code, model, data, and demo are available at: https://gorilla.cs.berkeley.edu

## 1   Introduction

The use of APIs and Large Language Models [10, 5, 31, 6, 27, 28] has changed what it means to program. Previously, building complex machine learning software and systems required extensive time and specialized skills. Now with tools like the HuggingFace API, an engineer can set up a deep learning pipeline with a few lines of code. Instead of searching through StackOverflow and documentation, developers can ask models like GPT for solutions and receive immediate, actionable code with docstrings. However, using off-the-shelf LLMs to generate API calls remains unsolved because there are millions of available APIs which are frequently updated.

We connect LLM's and massive API's with Gorilla, a system which takes an instruction, for example "build me a classifier for medical images", and provides the corresponding API call and relevant packages, along with a step-by-step explanation of the pipeline. Gorilla uses self-instruct, fine-tuning, and retrieval to enable LLMs to accurately select from a large, overlapping, and changing set tools expressed using their APIs and API documentation. Further, our novel retriever-aware training (RAT) enables the model to adapt to test-time changes of APIs such as evolution in versions and arguments.

With the development of API generation methods comes a question of how to evaluate, as many APIs will have overlapping functionality with nuanced limitations and constraints. Thus, we construct

38th Conference on Neural Information Processing Systems (NeurIPS 2024).

"Help me find an API to convert the spoken language in a recorded audio to text using Torch Hub."

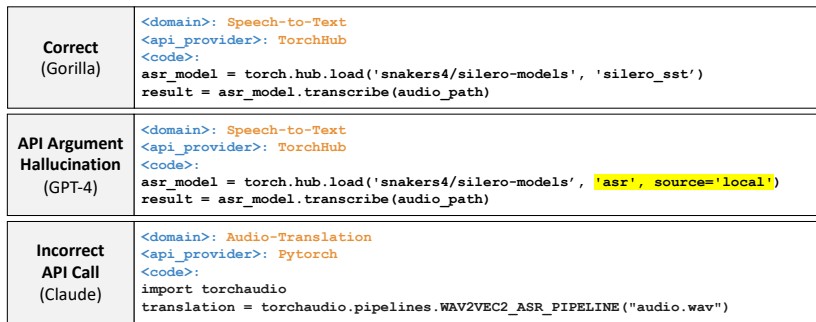

Figure 1: **Examples of API calls**. Example API calls generated by GPT-4 [27], Claude [2], and Gorilla for the given prompt. In this example, GPT-4 presents a model that doesn't exist, and Claude picks an incorrect library. In contrast, our Gorilla model can identify the task correctly and suggest a fully-qualified API call.

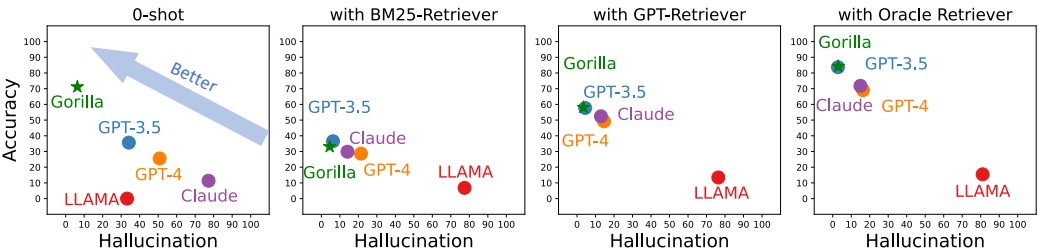

Figure 2: **Accuracy (vs) hallucination** in four settings, that is, *zero-shot* (i.e., without any retriever), and *with retrievers*. Commonly used BM25 and GPT retrievers, and the oracle – returns relevant documents with perfect recall, indicating an upper bound. Higher in the graph (higher accuracy) and to the left (lower hallucination) is better. Across settings, our model, Gorilla, improves accuracy while reducing hallucination.

APIBench ($\sim 1600$ APIs) by scraping a large corpus of ML APIs and developing an evaluation framework that uses AST sub-tree matching to check functional correctness. Further, we draw a distinction between accuracy and hallucination, and propose an Abstract Syntax Tree (AST) based technique to measure hallucination.

Using APIBench, we finetune Gorilla, a LLaMA-7B-based model with document retrieval and show that it significantly outperforms both open-source and closed-source models like Claude and GPT-4 in terms of API functionality accuracy as well as a reduction in API argument hallucination errors. We show an example output in Fig. 1. Lastly, we highlight Gorilla's capability to comprehend and reason about user-defined constraints when choosing between APIs, an essential requirement for LLMs trained to accomplish tasks.

To summarize, this paper makes the following contributions:

1. We introduce Gorilla, the first system to enable large-scale API integration with LLMs, demonstrating state-of-the-art performance in generating accurate API calls across thousands of functions and libraries.

2. We develop Retriever-Aware Training (RAT), a novel technique that enables LLMs to effectively utilize retrieved API documentation at inference time, improving both accuracy and adaptation to API changes.

3. We present APIBench, a comprehensive benchmark of $\sim 1600$ machine learning APIs, along with new AST-based evaluation metrics that precisely measure both functional correctness and API hallucination.

## 2   Related Work

By empowering LLMs to use tools [33], we can grant LLMs access to vastly larger and changing knowledge bases and accomplish complex computational tasks. By providing access to search technologies and databases, [24, 39, 35] demonstrated that we can augment LLMs to address a significantly larger and more dynamic knowledge space. Similarly, by providing access to computational tools, [39, 1, 49, 36, 37] demonstrated that LLMs can accomplish complex computational tasks. Consequently, leading LLM providers [27], have started to integrate plugins to allow LLMs to invoke external tools through APIs.

**Large Language Models**   Recent strides in the field of LLMs have renovated many downstream domains [10, 40, 48, 47], not only in traditional natural language processing tasks but also in program synthesis. Many of these advances are achieved by augmenting pre-trained LLMs by prompting [44, 14] and instruction fine-tuning [11, 30, 43, 15]. Recent open-sourced models like LLaMa [40], Alpaca [38], and Vicuna [9] have furthered the understanding of LLMs and facilitated their experimentation. While our approach, Gorilla, incorporates techniques akin to those mentioned, its primary emphasis is on enhancing the LLMs' ability to utilize millions of tools, as opposed to refining their conversational skills. Additionally, we pioneer the study of fine-tuning a base model by supplementing it with information retrieval - a first, to the best of our knowledge.

**Tool Usage**   The discussion of tool usage within LLMs has seen an upsurge, with models like Toolformer taking the lead [33, 19, 20, 24]. Tools often incorporated include web-browsing [32], calculators [12, 39], translation systems [39], and Python interpreters [14]. While these efforts can be seen as preliminary explorations of marrying LLMs with tool usage, they generally focus on specific tools. Our paper, in contrast, aims to explore a vast array of tools (i.e., API calls) in an open-ended fashion, potentially covering a wide range of applications.

With the recent launch of Toolformer [33] highlights the exciting potential of using large language models (LLMs) for purposes beyond traditional chatbot applications. Moreover, the application of API calls in robotics has been explored to some extent [41, 4]. However, these works primarily aim at showcasing the potential of "prompting" LLMs rather than establishing a systematic method for evaluation and training (including fine-tuning). Our work, on the other hand, concentrates on systematic evaluation and building a pipeline for future use.

**LLMs for Program Synthesis**   Harnessing LLMs for program synthesis has historically been a challenging task [22, 7, 45, 16, 13, 29]. Researchers have proposed an array of strategies to prompt LLMs to perform better in coding tasks, including in-context learning [44, 18, 7], task decomposition [17, 46], and self-debugging [8, 34]. Besides prompting, there have also been efforts to pretrain language models specifically for code generation [25, 21, 26].

DocPrompting [49] looked at choosing the right subset of code including API along with a retriever. Gorilla presents distinct advancements over DocPrompting. First, the way the data-sets are constructed are different, leading to intersting downstream artifacts. Gorilla focuses on model usages where we also collect detailed information about parameters, performance, efficiency, etc. This helps our trained model understand and respond to finer constraints for each API. Docprompting focuses on generic API calls but not on the details within an API call. Second, Gorilla introduces and uses the AST subtree-matching evaluation metric that helps measure hallucination which we find are more representative of code structure and API accuracy compared to traiditional NLP metrics. Finally, Gorilla focuses on instruction-tuning method and has "agency" to interact with users while DocPrompting focuses on building an NLP-to-Code generative model. On equal footing, we demonstrate that Gorilla performs better than DocPrompting in Appendix  A.3.

## 3   Methodology

We first describe APIBench, a comprehensive benchmark constructed from TorchHub, TensorHub, and HuggingFace API Model Cards. We begin by outlining the process of collecting the API dataset and how we generated instruction-answer pairs. We then introduce Gorilla, a novel training paradigm with an information–retriever incorporated into the training and inference pipelines. Finally, we present our AST tree matching evaluation metric.

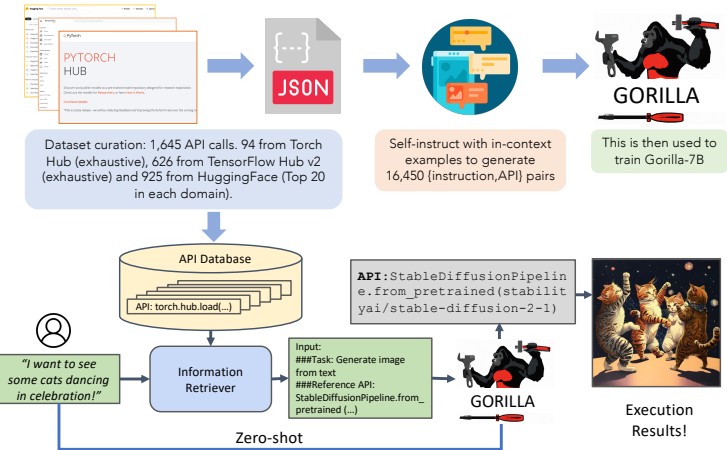

Figure 3: **Gorilla: A system for enabling LLMs to interact with APIs.** The upper half represents the training procedure as described in Sec 3. This is the most exhaustive API data-set for ML to the best of our knowledge. During inference (lower half), Gorilla supports two modes - with retrieval, and zero-shot. In this example, it is able to suggest the right API call for generating the image from the user's natural language query.

## 3.1  Dataset Curation

To curate the dataset, we aggregate all model cards from HuggingFace's "The Model Hub", PyTorch Hub, and TensorFlow Hub. Throughout the rest of the paper, we call these HuggingFace, Torch Hub, and TensorFlow Hub respectively for brevity.

**API Documentation**    The HuggingFace platform hosts and servers about 203,681 models. However, many of them have poor documentation, lack dependencies, have no information in their model card, etc. To filter these out, we pick the top 20 models from each domain. We consider 7 domains in multimodal data, 8 in CV, 12 in NLP, 5 in Audio, 2 in tabular data, and 2 in reinforcement learning. Post filtering, we arrive at a total of 925 models from HuggingFace. TensorFlow Hub is versioned into v1 and v2. The latest version (v2) has 801 models in total, and we process all of them. After filtering out model cards with little to no information, we are left with 626 models. Similar to TensorFlow Hub, we extract 95 models (exhaustive) from Torch Hub. We then convert the model cards for each of these 1,645 API calls into a JSON object with the following fields: {domain, framework, functionality, api_name, api_call, api_arguments, environment_requirements, example_code, performance, description}. We provide more information in Appendix A.1. These fields were chosen to generalize beyond API calls within the ML domain, to other domains, including RESTful, SQL, and other potential API calls.

**Instruction Generation**    Guided by the self-instruct paradigm [42], we employ GPT-4 to generate synthetic instruction data. We provide three in-context examples, along with reference API documentation, and task the model with generating real-world use cases that call upon the API. We specifically instruct the model to refrain from using any API names or hints when creating instructions. We constructed 6 examples (Instruction-API pairs) for each of the 3 model hubs. These 18 examples were the only hand-generated or modified data. For each of our 1,645 API datapoints, we generate 10 instruction-API pairs by sampling 3 of 6 corresponding instruction examples in each pair (Fig. 3).

**API Call with Constraints**    API calls often come with inherent constraints. These constraints necessitate that the LLM not only comprehend the functionality of the API call but also categorize the calls according to different constraint parameters. Specifically, for machine learning API calls, two common sets of constraints are parameter size and a lower bound on accuracy. Consider, for instance, the following prompt: *"Invoke an image classification model that uses less than 10M parameters, but maintains an ImageNet accuracy of at least 70%."* Such a prompt presents a substantial challenge for the LLM to accurately interpret and respond to. Not only must the LLM understand the user's functional description, but it also needs to reason about the various constraints embedded within the request. This challenge underlines the intricate demands placed on LLMs in real-world API calls. It is not sufficient for the model to merely comprehend the basic functionality of an API call; it must

also be capable of navigating the complex landscape of constraints that accompany such calls. We also incorporate these instructions in our training dataset.

## 3.2 Gorilla

Our model, called Gorilla, is a retriever-aware finetuned LLaMA-7B model, specifically for API calls. As shown in Fig. 3, we employ self-instruct to generate {instruction, API} pairs. To fine-tune LLaMA, we convert this to a user-agent chat-style conversation, where each datapoint is a conversation with one round each for the user and the agent. We then perform standard instruction finetuning on the base LLaMA-7B model. For our experiments, we train Gorilla with and without the retriever. We would like to highlight that though we used the LLaMA model, our fine-tuning is robust to the underlying pre-trained model (see Appendinx A.3.5).

**Retriever-Aware training (RAT)**   In retriever-aware training, the instruction-tuned dataset also appends to the user prompt, the relevant retrieved documentation with ''`Use this API documentation for reference: <retrieved_API_doc_JSON>`''. This is critical, because the retrieved documentation is not necessarily accurate – retrievers have imperfect re-call. By augmenting the prompt with potentially incorrect documentation, but the accurate ground-truth in the LLM response, we are in-effect teaching the LLM to 'judge' the retriever at inference time. During inference, if the LLM reasons that the retriever presented a relevant API document, it can use the API documentation to respond to the user's question, filling in additional details from the user's prompt. However, if after looking at the prompt, the LLM reasons that the retrieved API document is not relevant to the user's prompt, RAT trains the model to not get distracted by irrelevant context. The LLM then relies on the domain-specific knowledge baked-in during RAT training, to provide the user with the relevant API. Through RAT, we aim to teach the LLM to parse the second half of the question (API documentation) to answer the first half (user's query). We demonstrate that this (1) makes the LLM adapt to test-time changes in API documentation, (2) improves performance from in-context learning, and (3) reduces hallucination error.

Surprisingly, we find that augmenting a LLM with retrieval, does not always lead to improved performance, and can at-times hurt performance. We share more insights along with details in Sec 4.

**Gorilla Inference**   During inference, the user provides the prompt in natural language (Fig. 3). This can be for a simple task (e.g., *"I would like to identify the objects in an image"*), or they can specify a vague goal, (e.g., *"I am going to the zoo, and would like to track animals"*). Gorilla, similar to training, can be used for inference in two modes: zero-shot and with retrieval. In the zero-shot setting, this prompt (with no additional prompt tuning) is fed to the Gorilla LLM model, which then returns the API call needed to accomplish the task or goal. In retrieval mode, the retriever (either of BM25 or GPT-Index) first retrieves the most up-to-date API documentation stored in the API Database. Before being sent to Gorilla, the API documentation is concatenated to the user prompt along with the message *"Use this API documentation for reference."* The output of Gorilla is an API to be invoked. Besides the concatenation as described, we do *no* further prompt tuning in our system. While we also implemented a system to execute these APIs, to help the user accomplish the goal, that is not a focus of this paper.

## 3.3 Verifying APIs

Inductive program synthesis, where a program is synthesized to satisfy test cases, has found success in several avenues [3, 23]. However, test cases fall short when evaluating API calls, as it is often hard to verify the semantic correctness of the code. For example, consider the task of classifying an image. There are over 40 different models that can be used for the task. Even if we were to narrow down to a single family of `densenet`, there are four different configurations possible. Hence, there exist multiple correct answers and it is hard to tell if the API being used is functionally equivalent to the reference API by unit tests. Thus, to evaluate the performance of our model, we compare their functional equivalence using the dataset we collected. To trace which API in the dataset is the LLM calling, we adopt the AST tree-matching strategy. Since we only consider one API call in this paper, checking if the AST of the candidate API call is a sub-tree of the reference API call reveals which API is being used in the dataset.

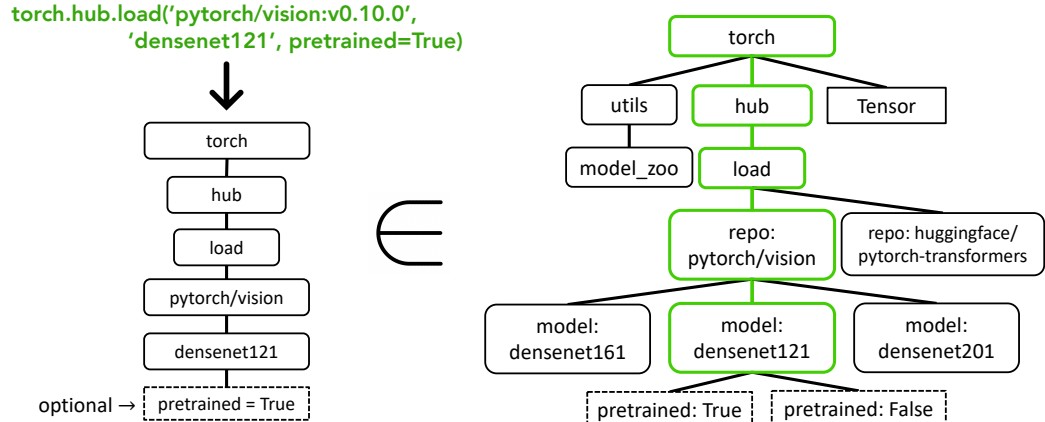

Figure 4: **AST Sub-Tree Matching to evaluate API calls.** On the left is an API call returned by Gorilla. We first build the associated API tree. We then compare this to our dataset, to see if the API dataset has a subtree match. In the above example, the matching subtree is highlighted in green, signifying that the API call is indeed correct. `Pretrained=True` is an optional argument.

Identifying and even defining hallucinations can be challenging. We use the AST matching process to directly identify the hallucinations. We define a hallucination as an API call that is not a sub-tree of any API in the database – invoking an entirely imagined tool. This form of hallucination is distinct from invoking an API incorrectly which we instead define as an error. So, in our evaluations, `error`, `hallucination`, and `accuracy` add up to one.

**AST Sub-Tree Matching**    We perform AST sub-tree matching to identify which API in our dataset is the LLM calling. Since each API call can have many arguments, we need to match on each of these arguments. Further, since, Python allows for default arguments, for each API, we define which arguments to match in our database. For example, we check `repo_or_dir` and `model` arguments in our function call. In this way, we can easily check if the argument matches the reference API or not. Fig. 4 illustrates an example subtree check for a torch API call. We first build the tree, and verify that it matches a subtree in our dataset along nodes `torch.hub.load`, `pytorch/vision`, and `densenet121`. We do not check for match along leaf node `pretrained=True` since that is an optional argument.

## 4   Evaluation

When evaluating Gorilla, finetuned on APIBench (*train set*), we aim to answer the following questions: How does Gorilla compare to other LLMs on API Bench (*test set*)? ( 4.1). How well does Gorilla adapt to test-time changes in API documentation? ( 4.2). How well can Gorilla handle questions with constraints? (4.3)

We demonstrate that Gorilla outperforms both open-source and close-source models for in-domain function calling. Further, trained with our novel retriever-aware training (RAT) technique, the Gorilla model generalizes to APIs that are outside of its training data (out-of-domain). In addition, we assess Gorilla's ability to reason about API calls under constraints. Lastly, we examined how integrating different retrieval methods during training influences the model's final performance.

**Baselines**    We primarily compare Gorilla with state-of-the-art language models in a zero-shot setting and with 3-shot in-context learning. The models under consideration include: GPT-4 by OpenAI with the `gpt-4-0314` checkpoint; GPT-3.5-turbo with the `gpt-3.5-turbo-0301` checkpoint, both of which are RLHF-tuned models specifically designed for conversation; Claude with the `claude-v1` checkpoint, a language model by Anthropic, renowned for its lengthy context capabilities; and LLaMA-7B, a state-of-the-art open-source large language model by Meta.

**Retrievers**    The term *zero-shot* (abbreviated as 0-shot in tables) refers to scenarios where no retriever is used. The sole input to the model is the user's natural language prompt. For BM25, we consider

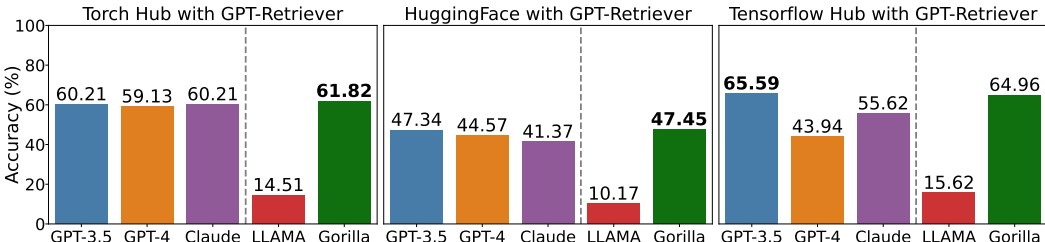

Figure 5: **Accuracy with GPT-retriever.** Methods to the left of the dotted line are closed source. Gorilla outperforms on Torch Hub and Hugging-Face while matching performance on Tensorflow Hub for all existing state-of-the-art LLMs - closed source, and open source.

each API as a separate document. During retrieval, we use the user's query to fetch the most relevant (top-1) API. This API is concatenated with the user's prompt to query the LLMs. Similarly, GPT-Index refers to the state-of-the-art embedding model, `text-embedding-ada-002-v2` from OpenAI, where each embedding is 1,536 dimensional. Like BM25, each API call is indexed as an individual document, and the most relevant document, given a user query, is retrieved and appended to the user prompt. Lastly, we include an Oracle retriever, which serves two purposes: first, to identify the potential for performance improvement through more efficient retrievers, and second, to assist users who know which API to use but may need to help invoking it. In all cases, when a retriever is used, it is appended to the user's prompt as follows: `<user_prompt>` Use this API documentation for reference: `<retrieved_API_doc_JSON>`. The dataset for these evaluations is detailed in Section 3. We emphasize that we have maintained a holdout test set on which we report our findings. The holdout test set was created by dividing the self-instruct dataset's instruction, API pairs into training and testing sets.

## 4.1 AST Accuracy on API call

We test each model for different retriever settings defined above (Table 1). We report the overall accuracy, the error by hallucination and the error by selecting wrong API call. Note that for TorchHub and TensorHub, we evaluate all the models using AST tree accuracy score. However, for HuggingFace, since the dataset cannot be exhaustive given the sheer number of models hosted, for all the models except Gorilla, we only check if they can provide the correct domain names. So this problem reduces to picking one of multiple choices. Across 0-shot and few-shot prompting strategies, Gorilla outperforms close-sourced and open-sourced models (Table 5).

**Finetuning without Retrieval** In Table 1 we show that lightly fine-tuned Gorilla is able to match, and often surpass performance in the zero-shot setting compared to closed-source, and open-source models – 20.43% better than GPT-4 and 10.75% better than GPT-3.5 (ChatGPT). When compared to other open-source models LLAMA, the improvement is as big as 83%. This suggests quantitatively, that as a technique to augment information and enforce adherence to syntax, fine-tuning is better than naive retrieval, at-least within the scope of invoking APIs.

**Finetuning with Retrieval** We now discuss how incorporating retrieval (RAT) during LLM fine-tuning enhances model performance. In this experiment, the base LLAMA model is finetuned with a prompt (instruction-generated), a reference API document (from a golden-truth oracle), and an example output generated by an LLM (GPT-4 in this case). As shown in Table 2, incorporating a ground-truth retriever in the finetuning pipeline yields notably improved results – 12.37% higher accuracy than training without retrieval in Torch Hub and 23.46% better in HuggingFace. However, at evaluation time, current retrievers show a significant performance gap compared to the ground-truth retriever: using GPT-Index at evaluation results in 29.20% accuracy degradation and using BM25 results in a 52.27% accuracy degradation. Despite this, considering the trends across models and retrievers, our findings indicate that finetuning an LLM with effective retrieval integration is preferable to zero-shot finetuning.

**Hallucination with LLM** One phenomenon we observe is that zero-shot prompting with LLMs (GPT-4/GPT-3.5) to call APIs results in dire hallucination errors. These errors,

Table 1: **Evaluating LLMs on Torch Hub, HuggingFace, and Tensorflow Hub APIs**

| LLM (retriever) | TorchHub | | | HuggingFace | | | TensorFlow Hub | | |
|---|---|---|---|---|---|---|---|---|---|
| | overall ↑ | hallu ↓ | err ↓ | overall ↑ | hallu ↓ | err ↓ | overall ↑ | hallu ↓ | err ↓ |
| LLAMA (0-shot) | 0 | 100 | 0 | 0.00 | 97.57 | 2.43 | 0 | 100 | 0 |
| GPT-3.5 (0-shot) | 48.38 | 18.81 | 32.79 | 16.81 | 35.73 | 47.46 | 41.75 | 47.88 | 10.36 |
| GPT-4 (0-shot) | 38.70 | 36.55 | 24.7 | 19.80 | 37.16 | 43.03 | 18.20 | 78.65 | 3.13 |
| Claude (0-shot) | 18.81 | 65.59 | 15.59 | 6.19 | 77.65 | 16.15 | 9.19 | 88.46 | 2.33 |
| Gorilla (0-shot) | **59.13** | **6.98** | 33.87 | **71.68** | **10.95** | 17.36 | **83.79** | **5.40** | 10.80 |
| LLAMA (BM-25) | 8.60 | 76.88 | 14.51 | 3.00 | 77.99 | 19.02 | 8.90 | 77.37 | 13.72 |
| GPT-3.5 (BM-25) | 38.17 | 6.98 | 54.83 | **17.26** | 8.30 | 74.44 | **54.16** | 3.64 | 42.18 |
| GPT-4 (BM-25) | 35.48 | 11.29 | 53.22 | 16.48 | 15.93 | 67.59 | 34.01 | 37.08 | 28.90 |
| Claude (BM-25) | 39.78 | 5.37 | 54.83 | 14.60 | 15.82 | 69.58 | 35.18 | 21.16 | 43.64 |
| Gorilla (BM-25) | **40.32** | **4.30** | 55.37 | 17.03 | **6.42** | 76.55 | 41.89 | **2.77** | 55.32 |
| LLAMA (GPT-Index) | 14.51 | 75.8 | 9.67 | 10.18 | 75.66 | 14.20 | 15.62 | 77.66 | 6.71 |
| GPT-3.5 (GPT-Index) | 60.21 | 1.61 | 38.17 | 29.08 | 7.85 | 44.80 | **65.59** | 3.79 | 30.50 |
| GPT-4 (GPT-Index) | 59.13 | 1.07 | 39.78 | 44.58 | 11.18 | 44.25 | 43.94 | 31.53 | 24.52 |
| Claude (GPT-Index) | 60.21 | 3.76 | 36.02 | 41.37 | 18.81 | 39.82 | 55.62 | 16.20 | 28.17 |
| Gorilla (GPT-Index) | **61.82** | **0** | 38.17 | **47.46** | **8.19** | 44.36 | 64.96 | **2.33** | 32.70 |
| LLAMA (Oracle) | 16.12 | 79.03 | 4.83 | 17.70 | 77.10 | 5.20 | 12.55 | 87.00 | 0.43 |
| GPT-3.5 (Oracle) | 66.31 | 1.60 | 32.08 | 89.71 | 6.64 | 3.65 | **95.03** | **0.29** | 4.67 |
| GPT-4 (Oracle) | 66.12 | 0.53 | 33.33 | 85.07 | 10.62 | 4.31 | 55.91 | 37.95 | 6.13 |
| Claude (Oracle) | 63.44 | 3.76 | 32.79 | 77.21 | 19.58 | 3.21 | 74.74 | 21.60 | 3.64 |
| Gorilla (Oracle) | **67.20** | **0** | 32.79 | **91.26** | **7.08** | 1.66 | 94.16 | 1.89 | 3.94 |

Table 2: **Understanding the effect of different retrieval techniques used with Gorilla**

| | Gorilla without Retriever | | | | Gorilla with Oracle retriever | | | |
|---|---|---|---|---|---|---|---|---|
| | zero-shot | BM25 | GPT-Index | Oracle | zero-shot | BM25 | GPT-Index | Oracle |
| Torch Hub (overall) ↑ | 59.13 | 37.63 | 60.21 | 54.83 | 0 | 40.32 | 61.82 | 67.20 |
| HuggingFace (overall) ↑ | 71.68 | 11.28 | 28.10 | 45.58 | 0 | 17.04 | 47.46 | 91.26 |
| TensorHub (overall) ↑ | 83.79 | 34.30 | 52.40 | 82.91 | 0 | 41.89 | 64.96 | 94.16 |
| Torch Hub (Hallu) ↓ | 6.98 | 11.29 | 4.30 | 15.59 | 100 | 4.30 | 0 | 0 |
| HuggingFace (Hallu) ↓ | 10.95 | 46.46 | 41.48 | 52.77 | 99.67 | 6.42 | 8.19 | 7.08 |
| TensorHub (Hallu) ↓ | 5.40 | 20.43 | 19.70 | 13.28 | 100 | 2.77 | 2.33 | 1.89 |

while diverse, commonly manifest in erroneous behavior such as the model invoking the `AutoModel.from_pretrained(dir_name)` command with arbitrary GitHub repository names. Surprisingly, we also found that in TorchHub, HuggingFace and TensorFlow Hub, GPT-3.5 has less hallucination errors than GPT-4. This finding is also consistent for the settings when various retrieving methods are provided: 0-shot, BM-25, GPT-Index and the oracle. This might suggest that RLHF plays a central role in turning the model to be truthful. Additional discussion in Appendix A.3.

**AST as a Hallucination Metric**   We manually execute Gorilla's API generations to evaluate how well AST works as an evaluation metric. Executing every code generated is impractical within academic setting—for example, executing the HuggingFace model needs the required library dependencies (e.g., transformers, sentencepiece, accelerate), correct coupling of software kernels (e.g., torch vision, torch, cuda, cudnn versions), and required hardware support (e.g., A100 40G gpus). Hence, to make it tractable, we sampled 100 random Gorilla generations from our evaluation set. The accuracy from our AST subtree matching is 78%, consistent with human evaluation of 78% accuracy in calling the right API. All the generations that AST flagged as incorrect, were the same ones that were manually flagged as incorrect. Additionally, Gorilla also generates supporting code to call the API which includes installing dependencies e.g., `pip install transformers[sentencepiece])`, setting environment variables, etc. When we manually attempt to execute the code, 72% of all code generated executed successfully. It's worth noting that the 6% discrepancy are not semantic errors, but errors that arose due to factors external to the API, and in the supporting code. We have included the full example to illustrate this further in A.3.3. Considering the significant time and effort required for manual validation of each generation, the strong correlation between human evaluation and the AST evaluation further reinforces our belief in using the proposed AST as a robust offline metric.

| Default Response | Update the model | Update the model repository |
|---|---|---|
| USER: I want to automatically remove the background from an input image. <Retrieval: ..fcn_resnet50..> | USER: I want to automatically remove the background from an input image <Retrieval: ..fcn_resnet101..> | USER: I want to automatically remove the background from an input image. <Retrieval: ..NVIDIA/DeepLearningExamples:torchhub..> |
| GORILLA: api_call: "torch.hub.load('pytorch/vision', 'fcn_resnet50, pretrained=True)" | GORILLA: api_call: "torch.hub.load('pytorch/vision', **'fcn_resnet101'**, pretrained=True)" | GORILLA: api_call: "torch.hub.load('**NVIDIA / DeepLearningExamples:torchhub** ', 'fcn_resnet50', pretrained=True)" |

Figure 6: **Gorilla's retriever–aware training enables it to react to changes in the APIs.** The second column demonstrates changes in model upgrading FCN's ResNet–50 backbone to ResNet-101. The third column demonstrate changes in model registry from `pytorch/vision` to `NVIDIA/DeepLearningExamples:torchhub`

Table 4: **Evaluating LLMs on constraint-aware API invocations**

|  | GPT-3.5 | | | | GPT-4 | | | | Gorilla | | | |
|---|---|---|---|---|---|---|---|---|---|---|---|---|
|  | 0-shot | BM25 | GPT-Index | Oracle | 0-shot | BM25 | GPT-Index | Oracle | 0-shot | BM25 | GPT-Index | Oracle |
| Torch Hub (overall) | **73.94** | 62.67 | 81.69 | 80.98 | 62.67 | 56.33 | 71.11 | 69.01 | 71.83 | 57.04 | 71.83 | 78.16 |
| Torch Hub (Hallu) | 19.01 | 30.98 | 14.78 | 14.08 | **15.49** | 27.46 | **14.08** | 9.15 | 19.71 | 39.43 | 26.05 | 16.90 |
| Torch Hub (err) | 7.04 | 6.33 | 3.52 | 4.92 | 21.83 | 16.19 | 14.78 | 21.83 | 8.45 | 3.52 | 2.11 | 4.92 |
| Accuracy const | 43.66 | **33.80** | **33.09** | 69.01 | 43.66 | 29.57 | 29.57 | 59.15 | **47.88** | 30.28 | 26.76 | 67.60 |

|  | LLAMA | | | | Claude | | | | |
|---|---|---|---|---|---|---|---|---|---|
|  | 0-shot | BM25 | GPT-Index | Oracle | 0-shot | BM25 | GPT-Index | Oracle | |
| Torch Hub (overall) | 0 | 8.45 | 11.97 | 19.71 | 29.92 | **81.69** | **82.39** | **81.69** | |
| Torch Hub (Hallu) | 100 | 91.54 | 88.02 | 78.87 | 67.25 | **16.19** | 15.49 | 13.38 | |
| Torch Hub (err) | 0 | 0 | 0 | 1.4 | 2.81 | 2.11 | 2.11 | 4.92 | |
| Accuracy const | 0 | 6.33 | 3.52 | 17.60 | 17.25 | 29.57 | 31.69 | **69.71** | |

Table 3: **Proposed AST evaluation metric has strong correlation with human evaluation**

|  | Accuracy |
|---|---|
| Gorilla AST metric (proposed) | 0.78 |
| Eval by Human | 0.78 |
| Code Executable (Eval by Human) | 0.72 |

## 4.2 Test-Time Documentation Change

The rapidly evolving nature of API documentation presents a significant challenge for the application of LLMs in this field. These documents are often updated at a frequency that outpaces the re-training or fine-tuning schedule of LLMs, making these models particularly brittle to changes in the information they are designed to process. This mismatch in update frequency can lead to a decline in the utility and reliability of LLMs over time.

With the introduction of Gorilla's retriever-aware training, the RAT trained LLM readily adapts to changes in API documentation. This novel approach allows the model to remain relevant, even as the API documentation it relies on undergoes modifications. This is a pivotal advancement in the field, as it ensures that the LLM maintains its efficacy and accuracy over time, providing reliable outputs irrespective of changes in the underlying documentation.

For instance, consider the scenario illustrated in Fig. 6, where the training of Gorilla has allowed it to react effectively to changes in APIs. This includes alterations such as upgrading the FCN's ResNet-50 backbone to ResNet-101, as demonstrated in the second column of the figure. Since the model has encountered ResNet-101 as a backbone with other architectures, it interprets an FCN with a ResNet-101 backbone (unseen during training) as a relevant document at test time. Conversely, if the retriever suggests an FCN with a ResNet-60 backbone, the model—unfamiliar with ResNet-60 from RAT—assigns low confidence to this document and defaults back to FCN with ResNet-50. The third column in Fig. 6 further illustrates Gorilla's flexibility in adapting to shifts in model registries, such as from `pytorch/vision` to `NVIDIA/DeepLearningExamples:torchhub`, highlighting its ability to accommodate changes in preferred API sources as they evolve over time.

Table 5: **Evaluating Gorilla 0-shot with GPT 3-shot incontext examples**

|  | HF (Acc ↑) | HF (Hall ↓) | TH (Acc ↑) | TH (Hall ↓) | TF (Acc ↑) | TF (Hall ↓) |
|---|---|---|---|---|---|---|
| GPT-3.5 (0-shot) | 16.81 | 35.73 | 41.93 | 10.75 | 41.75 | 47.88 |
| GPT-4 (0-shot) | 19.80 | 37.16 | 54.30 | 34.40 | 18.20 | 78.65 |
| GPT-3.5 (3 incont) | 25.77 | 32.30 | 73.11 | 72.58 | 71.82 | 11.09 |
| GPT-4 (3 incont) | 26.32 | 35.84 | **75.80** | **13.44** | 77.37 | 11.97 |
| Gorilla (0-shot) | **58.05** | **28.32** | **75.80** | 16.12 | **83.79** | **5.40** |

In summary, Gorilla's ability to adapt to test-time changes in API documentation offers numerous benefits. It maintains its accuracy and relevance over time, adapts to the rapid pace of updates in API documentation, and adjusts to modifications in underlying models and systems. This makes it a robust and reliable tool for API calls, significantly enhancing its practical utility.

### 4.3 API Call with Constraints

We now focus on the language model's capability of understanding constraints. For any given task, which API call to invoke is typically a tradeoff between a multitude of factors. In the case of RESTFul APIs, it could be the cost of each invocation ($) or the latency of response (ms), among many others. Similarly, within the scope of ML APIs, it is desirable for Gorilla to respect constraints such as accuracy, number of learnable parameters in the model, the size on disk, peak memory consumption, FLOPS, etc. In this section, we present a study evaluating the ability of different models in zero-shot and in the presence of retrievers to respect a given accuracy constraint. : if a user requests an image classification model that achieves at least 80% top-1 accuracy on the ImageNet dataset, then among the classification models hosted by Torch Hub, `ResNeXt-101 32x16d`, with a top-1 accuracy of 84.2%, would be the appropriate model to call, rather than `MobileNetV2`, which has a top-1 accuracy of 71.88%.

For Table 4, we filtered a subset of the Torch Hub component of APIBench, retaining those models that had an accuracy metric defined for at least one-dataset the model was evaluated on, in its model card. We were left with 65.26% of TorchHub dataset from Table 1. We notice that with constraints, understandably, the accuracy drops across all models, with and without a retriever. Even in this challenging scenario, Gorilla is able to match the performance of the best-performing model GPT-3.5 when using retrievals (BM25, GPT-Index), and has the highest accuracy in the zero-shot setting. This highlights Gorilla's ability to navigate APIs while considering the trade-offs between constraints.

### 4.4 Finetuning (vs) Prompting: Gorilla 0-shot (vs) GPT 3-shot

To assess whether finetuning is truly necessary for APIs or if prompting alone is sufficient, we compare Gorilla in a zero-shot setting with three-shot in-context prompting for GPT-3.5 and GPT-4 models. In Table 5, "3-incont" denotes evaluation using three in-context examples, while "HF," "TH," and "TF" represent the HuggingFace, TorchHub, and TensorFlow Hub subsets of APIBench, respectively. Higher accuracy (Acc) and lower hallucination (Hall) rates are preferred. From Table 5, three-shot in-context learning improves the GPT models' ability to generate syntactically correct function calls, even matching accuracy on one subset (TorchHub). However, Gorilla 0-shot still outperforms the 3-shot GPT models on average.

## 5 Conclusion

LLMs are swiftly gaining popularity across diverse domains. APIs, serving as a universal language, are essential for enabling LLMs to communicate and operate effectively across diverse systems. In this paper, we introduced Gorilla, a state-of-the-art model for API invocation. Our Retriever Aware Training (RAT) approach empowers Gorilla with two essential capabilities: adapting dynamically to API changes at test time and reasoning through user-defined constraints when selecting suitable APIs. We also present APIBench, a comprehensive benchmark for assessing LLMs' function-calling abilities, and propose AST-based hallucination metrics for robust evaluation. Looking forward, we believe this work represents a first step towards transitioning LLMs from knowledge-bound models into flexible interfaces that interact with the digital world.

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

# A Appendix

## A.1 Dataset Details

Our dataset is multi-faceted, comprising three distinct domains: Torch Hub, Tensor Hub, and HuggingFace. Each entry within this dataset is rich in detail, carrying critical pieces of information that further illuminate the nature of the data. Delving deeper into the specifics of each domain, Torch Hub provides 95 APIs. The second domain, Tensor Hub, is more expansive with a total of 696 APIs. Finally, the most extensive of them all, HuggingFace, comprises 925 APIs.

To enhance the value and utility of our dataset, we've undertaken an additional initiative. With each API, we have generated a set of 10 unique instructions. These instructions, carefully crafted and meticulously tailored, serve as a guide for both training and evaluation. This initiative ensures that every API is not just represented in our dataset, but is also comprehensively understood and effectively utilizable.

In essence, our dataset is more than just a collection of APIs across three domains. It is a comprehensive resource, carefully structured and enriched with added layers of guidance and evaluation parameters.

**Domain Classification**  The unique domain names encompassed within our dataset are illustrated in Fig. 7. The dataset consists of three sources with a diverse range of domains: Torch Hub houses 6 domains, Tensor Hub accommodates a much broader selection with 57 domains, while HuggingFace incorporates 37 domains. To exemplify the structure and nature of our dataset, we invite you to refer to the domain names represented in Fig. 8.

**API Call Task**  In this task, we test the model's capability to generate a single line of code, either in a zero-shot fashion or by leveraging an API reference. Primarily designed for evaluation purposes, this task effectively gauges the model's proficiency in identifying and utilizing the appropriate API call.

**API Provider Component**  This facet relates to the provision of the programming language. In this context, the API provider plays a vital role as it serves as a foundation upon which APIs are built and executed.

**Explanation Element**  This component offers valuable insights into the rationale behind the usage of a particular API, detailing how it aligns with the prescribed requirements. Furthermore, when certain constraints are imposed, this segment also incorporates those limitations. Thus, the explanation element serves a dual purpose, offering a deep understanding of API selection, as well as the constraints that might influence such a selection. This balanced approach ensures a comprehensive understanding of the API usage within the given context.

**Code**  Example code for accomplishing the task. We de-prioritize this as we haven't tested the execution result of the code. We leave this for future works, but make this data available in-case others want to build on it.

## A.2 Gorilla Details

We provide all the training details for Gorilla in this section. This includes how we divide up the training, evaluation dataset, training hyperparameters for Gorilla.

**Data**  For HuggingFace, we devise the entire dataset into 90% training and 10% evaluation. For Torch Hub and Tensor Hub, we devise the data in to 80% training and 20% testing.

**Training**  We train Gorillafor 5 epochs with the 2e-5 learning rate with cosine decay. The details are provide in Table 6. We finetune it on 8xA100 with 40G memory each.

**Torch Hub domain names**: Classification, Semantic Segmentation, Object Detection, Audio Separation, Video Classification, Text-to-Speech

**Tensor Hub domain names**: text-sequence-alignment, text-embedding, text-language-model, text-preprocessing, text-classification, text-generation, text-question-answering, text-retrieval-question-answering, text-segmentation, text-to-mel, image-classification, image-feature-vector, image-object-detection, image-segmentation, image-generator, image-pose-detection, image-rnn-agent, image-augmentation, image-classifier, image-style-transfer, image-aesthetic-quality, image-depth-estimation, image-super-resolution, image-deblurring, image-extrapolation, image-text-recognition, image-dehazing, image-deraining, image-enhancemenmt, image-classification-logits, image-frame-interpolation, image-text-detection, image-denoising, image-others, video-classification, video-feature-extraction, video-generation, video-audio-text, video-text, audio-embedding, audio-event-classification, audio-command-detection, audio-paralinguists-classification, audio-speech-to-text, audio-speech-synthesis, audio-synthesis, audio-pitch-extraction

**HuggingFace domain names**: Multimodal Feature Extraction, Multimodal Text-to-Image, Multimodal Image-to-Text, Multimodal Text-to-Video, Multimodal Visual Question Answering, Multimodal Document Question Answer, Multimodal Graph Machine Learning, Computer Vision Depth Estimation, Computer Vision Image Classification, Computer Vision Object Detection, Computer Vision Image Segmentation, Computer Vision Image-to-Image, Computer Vision Unconditional Image Generation, Computer Vision Video Classification, Computer Vision Zero-Shor Image Classification, Natural Language Processing Text Classification, Natural Language Processing Token Classification, Natural Language Processing Table Question Answering, Natural Language Processing Question Answering, Natural Language Processing Zero-Shot Classification, Natural Language Processing Translation, Natural Language Processing Summarization, Natural Language Processing Conversational, Natural Language Processing Text Generation, Natural Language Processing Fill-Mask, Natural Language Processing Text2Text Generation, Natural Language Processing Sentence Similarity, Audio Text-to-Speech, Audio Automatic Speech Recognition, Audio Audio-to-Audio, Audio Audio Classification, Audio Voice Activity Detection, Tabular Tabular Classification, Tabular Tabular Regression, Reinforcement Learning Reinforcement Learning, Reinforcement Learning Robotics

Figure 7: **Domain names**: Domain names with the three dataset. Tensor Hub is the smallest dataset while the other two hubs contain many more models.

Table 6: Hyperparameters for training Gorilla

| Hyperparameter Name | Value |
| --- | --- |
| learning rate | 2e-5 |
| batch size | 64 |
| epochs | 5 |
| warmup ratio | 0.03 |
| weight decay | 0 |
| max seq length | 2048 |

### A.3 Performance Comparison

We provide a full comparison of each model's performance in this section. In Fig 10 and Fig. 11, the full set of comparisons is provided. We see that especially in zero-shot case, Gorilla surpasses the GPT-4 and GPT-3.5 by a large margin. The GPT-4 and GPT-3.5 gets around 40% accuracy in Torch Hub and Tensor Hub, which are two structured API calls. Compared to that, HuggingFace is a more flexible and diverse Hub, as a result, the performance on HuggingFace is not as competitive.

### User: Our customer is a zoo and we want to help them detect movement of different animals. Write a python program in 1 to 2 lines to call API in TensorFlowHub.
### Assistant:
**<domain>**: image-object-detection,
**<api_call>**: detector = hub.load('https://tfhub.dev/google/openimages_v4/ssd_mobilenet_v2/1'),
**<api_provider>**: TensorFlow Hub,
**<explanation>**: This API call loads the pre-trained Single Shot Detector (SSD) MobileNet V2 model from TensorFlow Hub, which has been trained on the Open Images v4 dataset. This model is used for object detection in images. Once loaded, you can pass an image (and an optional score threshold for detection confidence) to the detector to get back the detected objects along with their bounding boxes, class labels, and associated confidence scores.
****: import tensorflow as tf; import tensorflow_hub as hub; # Load the SSD MobileNet V2 model; detector = hub.load('https://tfhub.dev/google/openimages_v4/ssd_mobilenet_v2/1'); # Use the model to detect objects in an image (assuming 'image' is a TensorFlow tensor representing your image); result = detector(image, score_threshold=0.5)

---

### User: I am an engineer at Uber and I need to find a API that can classify pedestrians, cars, etc. from an image of the scene. Write a python program in 1 to 2 lines to call API in TorchHub.
**<Reference API>**: {domain: Object Detection, framework: PyTorch, functionality: Traffic Object Detection, Drivable Area Segmentation, Lane Detection, api_name: HybridNets, api_call: torch.hub.load(repo_or_dir='datvuthanh/hybridnets', model='hybridnets', pretrained=True), api_arguments: repo_or_dir, model, pretrained, python_environment_requirements: Python>=3.7, PyTorch>=1.10, example_code: 'import torch; model = torch.hub.load('datvuthanh/hybridnets', 'hybridnets', pretrained=True); img = torch.randn(1,3,640,384); features, regression, classification, anchors, segmentation = model(img), performance: dataset: [name: BDD100K, accuracy: Traffic Object Detection: Recall (%): 92.8, mAP@0.5 (%): 77.3, Drivable Area Segmentation: Drivable mIoU (%): 90.5, Lane Line Detection: Accuracy (%): 85.4, Lane Line IoU (%): 31.6], description: HybridNets is an end2end perception network for multi-tasks. Our work focused on traffic object detection, drivable area segmentation and lane detection. HybridNets can run real-time on embedded systems, and obtains SOTA Object Detection, Lane Detection on BDD100K Dataset.}
### Assistant:
**<domain>**: Object Detection, **<api_call>**: model = torch.hub.load('datvuthanh/hybridnets', 'hybridnets', pretrained=True),
**<api_provider>**: PyTorch,
**<explanation>**: Load the pretrained HybridNets model for object detection from PyTorch Hub to classify pedestrians, cars, etc. in an image of the scene,
****: import torch; model = torch.hub.load('datvuthanh/hybridnets', 'hybridnets', pretrained=True)

Figure 8: **Example of the Dataset**: Two examples of the dataset, the above one is zero-shot (without information retrievers) and the bottom one is with information retriever.

```
generate_video = pipeline("text-to-video", model="your_model_name")
```

```
vqa = pipeline("visual-question-answering", model="microsoft/clip-vqa-base",
tokenizer="microsoft/clip-vqa-base")
```

```
depth_estimator = pipeline("depth-estimation", model="intel-isl/MiDaS", tokenizer="intel-
isl/MiDaS")
```

Figure 9: **Hallucination Examples**: GPT-4 incurs serious hallucination errors in HuggingFace. We show a couple of examples in the figure.

### A.3.1 Evaluation

For ease of evaluation, we manually cleaned up the dataset to ensure each API domain only contains the valid call of form:

$$\text{API\_name(API\_arg}_1\text{, API\_arg}_2\text{, ..., API\_arg}_k\text{)}$$

Our framework allows the user to define any combination of the arguments to check. For Torch Hub, we check for the API name `torch.hub.load` with arguments `repo_or_dir` and `model`. For Tensor Hub, we check API name `hub.KerasLayer` and `hub.load` with argument `handle`. For HuggingFace, since there are many API function names, we don't list all of them here. One specific note is that we require the `pretrained_model_name_or_path` argument for all the calls except for `pipeline`. For `pipeline`, we don't require the `pretrained_model_name_or_path` argument since it automatically select a model for you once `task` is specified.

### A.3.2 Hallucination

We found especially in HuggingFace, the GPT-4 model incurs serious hallucination problems. It would sometimes put a GitHub name that is not associated with the HuggingFace repository in to the domain of `pretrained_model_name_or_path`. Fig. 9 demonstrates some examples and we also observe that GPT-4 sometimes assumes the user have a local path to the model like `your_model_name`. This is greatly reduced by Gorilla as we see the hallucination error comparison in Table 1.

### A.3.3 AST as a Hallucination Metric

We evaluated the generated results on 100 LLM generations (randomly chosen from our eval set). The accuracy using AST subtree matching is 78%, consistent with human evaluation with 78% accuracy in calling the right API. All the generations that AST flagged as incorrect, were the same ones that were manually also flagged as incorrect. Additionally, Gorilla generates supporting code to call the API which includes installing dependencies (e.g., `pip install transformers[sentencepiece]`), environment variables, etc. When we manually attempted to execute end-to-end code, 72% of all codes generated were executed successfully. It's worth noting that the 6% discrepancy were NOT semantic errors, but errors that arose due to factors external to the API in the supporting code - we have included an example to illustrate this further. Considering the significant time and effort required for manual validation of each generation, our evaluation highlights the efficiency of using AST as a robust offline metric.

Here is a representative example, where we are able to load the correct model API. However, in the supporting code, after we have the output from the API, the `zip()` function tries to combine sentiments and scores together. However, since scores is a float, it's not iterable. `zip()` expects both its arguments to be iterable, resulting in an `'float' object is not iterable` error.

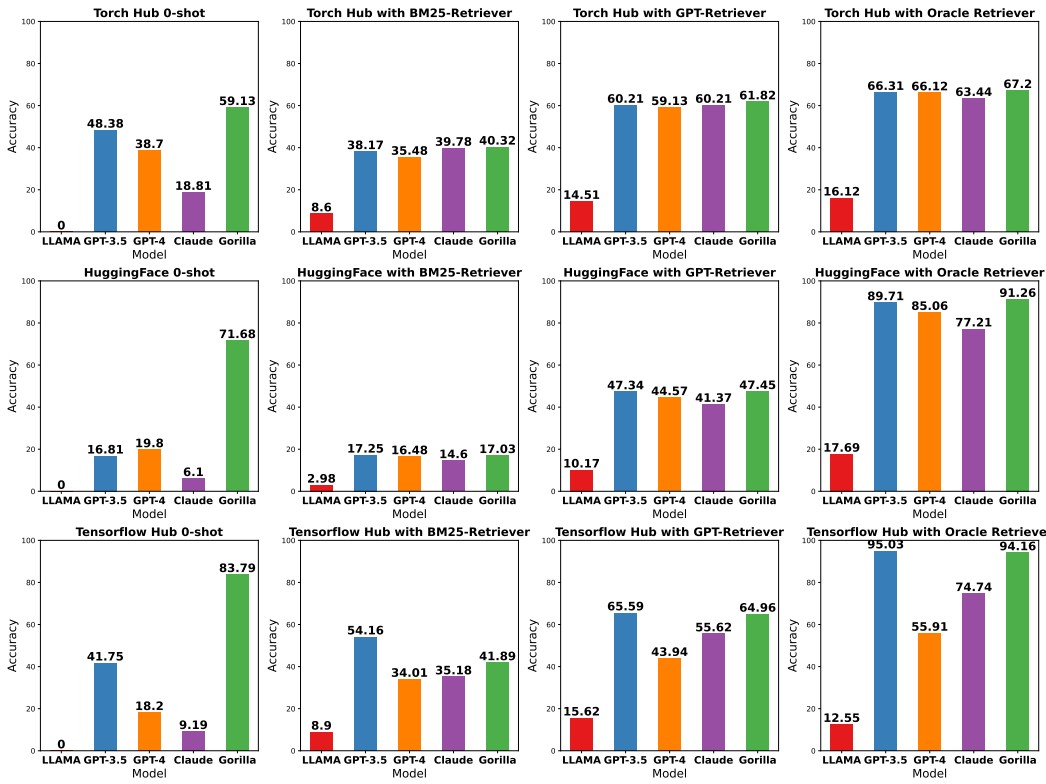

Figure 10: **Performance**: We plot each model's performance on different configurations. We see that Gorilla performs extremely well in the zero-shot setting. While even when the oracle answer is given, Gorilla is still the best.

Table 7: **Evaluating Gorilla (vs) DocPrompting** Gorilla improves accuracy, while lowering the hallucination.

| Accuracy ↑ | | Hallucination ↓ | |
|---|---|---|---|
| DocPrompting | Gorilla | DocPrompting | Gorilla |
| 61.72 | **71.68** | 17.36 | **10.95** |

### A.3.4  Gorilla (VS) DocPrompting

We evaluate Gorilla and DocPrompting [49] on the HuggingFace Dataset from Table 1. For a 7B model, when trained on the same number of epochs, with and the same learning rate for both the models, Gorilla improves accuracy while reducing hallucination.

### A.3.5  Sensitivity to pre-training

Gorilla's training recipe is robust to the pre-training strategies and recipes of the underlying model. From Fig. 13 we demonstrate that all the three models can converge to within a few percentage points in accuracy independent of the pre-trained base model.

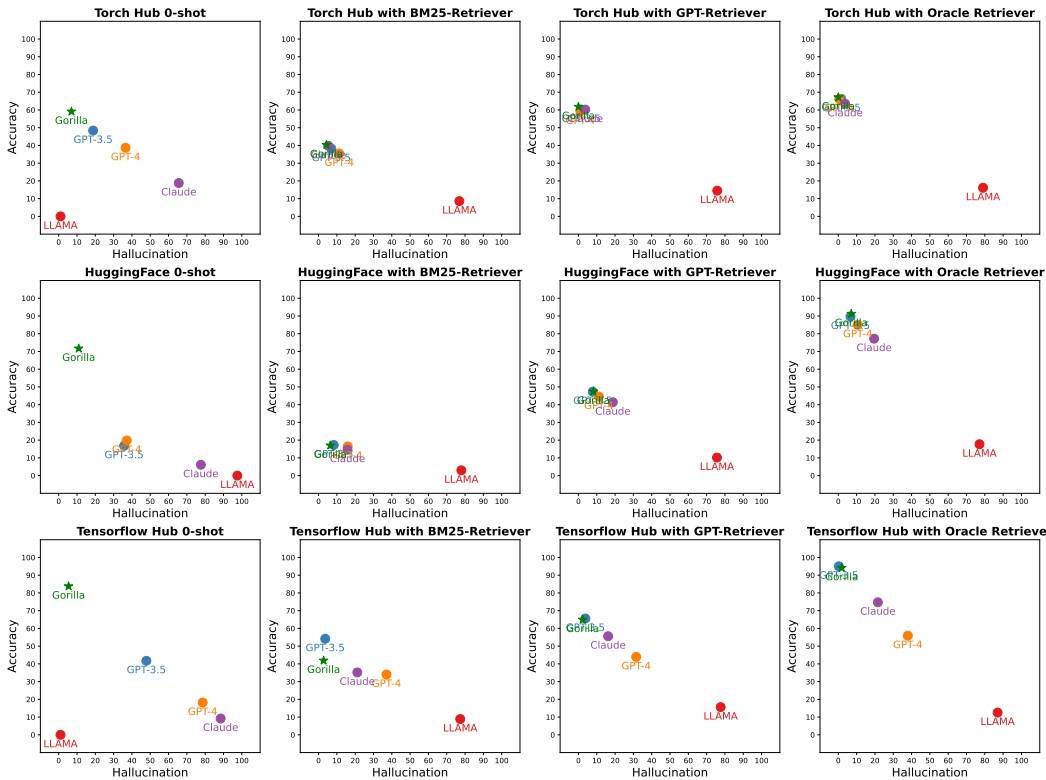

Figure 11: **Accuracy vs Hallucination**: We plot each model's performance on different configurations. We found that in the zero-shot setting, Gorilla has the most accuracy gain while maintaining good factual capability. When prompting with different retrievers, Gorilla is still capable to avoid the hallucination errors.

```python
from transformers import pipeline

def load_model():
    classifier = pipeline('sentiment-analysis',
        model='nlptown/bert-base-multilingual-uncased-sentiment')
    return classifier

def process_data(comments, classifier):
    response = classifier(comments)
    sentiments = response[0]['label'].split()
    scores = response[0]['score']
    result = [{'sentiment': sentiment, 'score': score}
        for sentiment, score in zip(sentiments, scores)]
    return result

comments = "These comments are about our news website."
# Load the model
classifier = load_model()
# Process the data
response = process_data(comments, classifier)
print(response)
```

Figure 12: The API call by Gorilla model are accurate and bug-free, but the supporting `zip()` code has a bug.

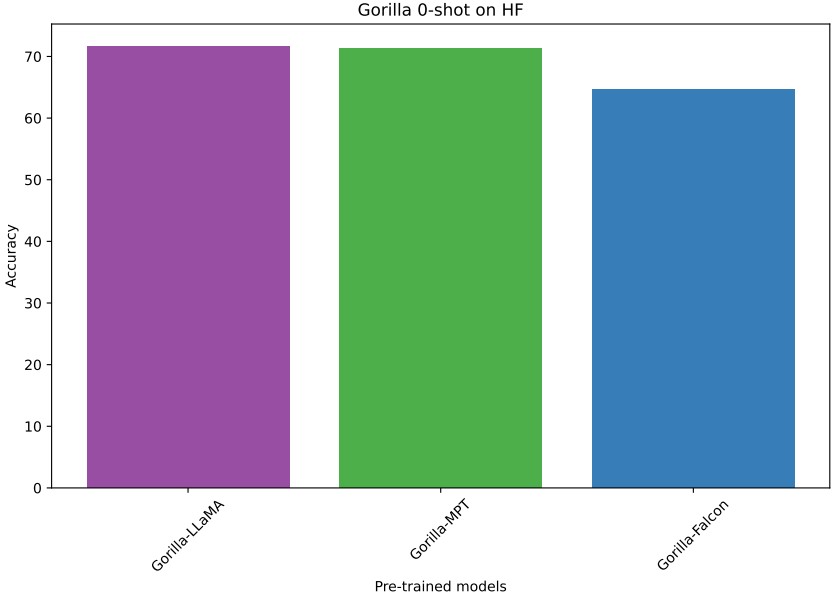

Figure 13: For the same train-eval dataset, our fine-tuning recipe, RAT, is robust to the underlying base model.

