# OpenReview forum: "Gorilla: Large Language Model Connected with Massive APIs"
_NeurIPS.cc/2024/Conference — NeurIPS 2024 poster_

### Official Review · Reviewer_N9sF · 2024-07-03

**Soundness:** 3
**Presentation:** 4
**Contribution:** 3
**Rating:** 6
**Confidence:** 3

**Summary:**

This paper proposes Gorilla, a fine-tuned LLaMA model, outperforms GPT-4 in crafting API calls and adapts well to document changes with a document retriever, reducing hallucination issues. It also provides APIBench, a new dataset for evaluation, includes APIs from HuggingFace, TorchHub, and TensorHub. Gorilla's integration with a retrieval system enhances tool usage accuracy and documentation updates, promising more reliable LLM outputs.

**Strengths:**

1. The paper trains a system which connects massive APIs and takes text instruction to get the corresponding API calls, along with a step-by-step explanation of the pipeline. It focuses on the details within an API call, which significantly mitigates the complexity involved in developing machine learning softwares.
2. The paper also constructs APIBench by scraping a large corpus of ML APIs and developing an evaluation framework to check functional correctness. It uses AST subtree-matching metrics which helps measure hallucination and further contributes to the evaluation of ML API mastering techniques.
3. The paper is well-writtened, presenting a detailed comparison with other related works, a clear structure on methods, and comprehensive experiments and analysis.

**Weaknesses:**

1. Although the paper overall does not present significant issues, I would appreciate seeing a performance comparison with more specific methods, such as other works that also utilize LLM for tool calling to build software pipelines, which may further demonstrate the contribution of techniques presented by Gorilla.
2. The paper needs to better illustrate the contributions of this work and more clearly outline its potential impact on the community.
3. Some writing detales need to be improved in the paper, such as "the" should be omitted in "The model trained with the our method" in Line 31, and "AST" should be replaced by "Abstract Syntax Tree" for the first time it appears in Line 36.

**Questions:**

1. In Figure 2, why GPT-3.5 is much better than GPT-4 in all settings?
2. As most of the instructions are generated by GPT-4 referring to your hand-generated instruction seeds. How to ensure your seeds meets the real-world scenario and reach the real difficulties?
3. As many APIs have overlapping functions, how to decide which API is better than the other similar one? How to maintain stable API calling to persist consistent results for the same request?
4. Which Llama version is used as your foundation model? Llama2 or Llama3?
5. In Table 1, why Gorilla always produces a large error in selecting wrong APIs? Do the results support your view?

**Limitations:**

The paper should provide a more comprehensive limitation part to point out potential overlooked issues, such as influence of synthetic training data and insufficient comparative experiments.

---

> ### Author Rebuttal · Authors · 2024-08-07
>
> We thank the reviewer for their time and thoughtful comments. We are encouraged you enjoyed the exposition of our pipeline and find the APIBench, and the AST subtree-matching metric to contribute to mastering techniques to evaluate APIs.
>
> **1. Highlighting the contributions and potential impact on community**
>
> Gorilla is centered around connecting LLMs with APIs, and to this end, we:
> 1. Develop and manually curate a high-quality dataset that is valuable to the community.
> 2.RAT: We introduce Retriever-Aware Training (RAT) which teaches the LLM to either use or ignore the retrieved context. This is a novel approach that helps Gorilla achieve superior performance.
> 3. Propose evaluation metrics, including the first technique to define and measure hallucination for the domain of (functions and tools) APIs. We also perform human evaluation to validate the competency of the metric.
> 4. Study realistic API calls with constraints (e.g., “I would like to detect pedestrians and cars on the street. I want a model to run on a Raspberry PI with accuracy at least 65% on ImageNet”. Here, Gorilla needs to reason about the constraints - that the model needs to fit the memory budget while still meeting accuracy budgets.)
> 5. Measure hallucination for popular models for APIs.
>
> **2. Comparison with other LLMs for tool calling**
>
> Beyond frontier labs whose models we have compared, within academic literature, the strongest baseline (API-focused model) when submitting the paper is DocPrompting Zhou et al. (ICLR 2023) which looked at choosing the right subset of code including API along with a retriever. We demonstrate that when comparing Gorilla (vs) DocPrompting, Gorilla improves accuracy, while lowering the hallucination. We are happy to include any new baselines you might suggest.
>
> |Accuracy ↑ | Accuracy ↑ | Hallucination ↓ | Hallucination ↓ |
> |---|---|---|---|
> |DocPrompting | **Gorilla** | DocPrompting | **Gorilla**|
> |61.72 |**71.68**| 17.36| **10.95**|
>
> We also include a comparison with **prompting techniques {0-shot} and {3-shot}** in Table 6 of Appendix, and demonstrate Gorilla improves performance across both techniques for GPT-3.5 and GPT-4.
>
> **3. Writing Detales**
>
> Thank you for flagging these. We agree that the writing mechanics of the paper can be improved and have already made editorial passes over the paper to improve the exposition of the paper. We will also include more comprehensive limitation part to point out potential issues, such as influence of synthetic training data and comparative experiments, and have already reflected the changes suggested.
>
> **4. Comparing GPT-3.5 vs GPT-4**
>
> We would first like to highlight that with frontier models we can’t know for sure. Further, perhaps, this is orthogonal to this work. However, this intrigued us as well. One potential explanation we got when we reached out to OpenAI is that GPT-4 is “lightly” RLHF’d when compared to GPT-3.5. Making 3.5 a stronger instruction follower. So, we can observe that as we move from 0-shot to oracle-retriever, 3.5 shines, since it is better at “following the instruction” of using the API to answer the user question, while GPT-4 on the other hand, tends to hallucinate even when provided with an API - oracle or not.
>
> **5. Diverse real-world scenarios for instruction**
>
> For the self-instruct generation, we provide three in-context examples, along with reference API documentation, and task the model with generating real-world use cases that call upon the API. We specifically instruct the model to refrain from using any API names or hints when creating instructions. We constructed 6 examples (Instruction-API pairs) for each of the 3 model hubs. Post this, we manually verify and audit each generation resulting in high quality data-set. Making this an important contribution to the community.
> Further, as a yard-stick Gorilla models have been downloaded 10000+ times.
> Here are some examples randomly chosen to demonstrate the diversity and versatility:
> ```
> Example 1 (q 638) “I am an illustrator, I want to create an appealing image based on a text description for commercial purposes.”
>
> Example 2 (q 704) “Assist me in finding the accurate information in a table related to the NYSE stock market”
>
> Example 3 (q 867)  “We want to communicate product information to online customers in France. 'Introducing the new eco-friendly water bottle made of high-quality stainless steel with double-wall insulation to keep your drinks cool for 24 hours or hot for 12 hours'.”
> ```
>
> **6. How to decide which API is better than the other similar one**
>
> Given a question, depending on the constraints we narrow down the set of right answers. For example, if the user requests an object detection model - any and all object detection APIs are the right answers. However, if a user specifies a constraint either explicitly or implicitly (e.g., they want the model to have a top-1 accuracy of xx on imagenet) then the set of APIs narrows down and a subset of them are “better” than other similar ones. Since, we create the data-set we can control for this. If two APIs are the right answer, then if either of them are called, we mark that as accurate.
>
> **7. In Table 1, why (does) Gorilla always produce a large error in selecting wrong APIs? Do the results support your view?**
>
> Yes, this is quite intuitive. By teaching the model domain knowledge (with our novel Retriever Aware Training), the model tends to hallucinate less (not 0, but lesser than baselines). The error results from the model unable to identify the user intent and generating API for the wrong task.
>
> **8. Which Llama version is used as your foundation model?**
>
> Llama-2 model. We evaluate the performance across three diverse base models : MPT-7B (0.70), Falcon-7B (0.66) and LLAMA-7B (0.71). The results are included in the paper (Appendix Figure 13) and demonstrate that all kept constant, our innovative Retrieval Aware Training (RAT) fine-tuning recipe is robust to the underlying base model.

---

> > ### Comment · Reviewer_N9sF · 2024-08-09
> > **Response to the author**
> >
> > Thank you for your reply. Your explanation has mostly addressed my concerns.
> >
> > I hope you can incorporate some of the above content into the revised paper, such as the contributions and impact to the community, comparisons with other methods based on LLMs for tool calling, and some grammar issues. However, since there are still some experimental phenomena whose causes have not been fully explained, I suggest the author conduct a more in-depth analysis and investigation. Therefore, I have decided to keep my scores.

---

### Official Review · Reviewer_jdAv · 2024-07-06

**Soundness:** 2
**Presentation:** 2
**Contribution:** 3
**Rating:** 6
**Confidence:** 4

**Summary:**

The authors present a new fine-tuned language model that is trained to map from user instructions to code snippets that invoke the appropriate APIs. The authors also introduce a new dataset which includes the information for roughly 1600 APIs from a variety of online sources, which is used to train the model. They conclude that their approach outperforms existing language models on the task of API call generation, as measured by both error rate and a novel “hallucination” metric.

**Strengths:**

I feel that the problem statement is well-motivated and of general interest. Prior work has investigated allowing LLMs to make use of various tools, and improvements in that capability are likely to be well received. I appreciate that the authors have chosen a (relatively speaking) lightweight and open-source model, which increases the usability of their approach. I also appreciate the construction of a novel dataset, which could presumably be of interest to the larger community even in the absence of a specific model.

**Weaknesses:**

First, I wonder about baselines. The authors have done a good job of comparing their method against a variety of open-source and closed-source LLMs, but these systems are all “generalist” compared to the fine-tuned Gorilla model. It would have been useful to see a comparison to a fine-tuned version of another model (even one with less fine-tuning than is possible on the small Llama-7b) just to have a sense of how much improvement can be leveraged from the newly introduced dataset. I also wonder if it would have been possible to compare to specialized tool-based LLM models like Toolformer. If such a comparison is not appropriate or possible, I feel the authors should mention why.

My second question concerns statistical significance. The authors indicate the high cost of LLM experiments as the reason for omitting such analysis. I sympathize with this explanation, but feel that I would be remiss not to stress the importance of statistical testing as part of justifying claims that Gorilla’s “performance surpasses prompting the state-of-the-art LLMs in three massive datasets.” Without a measure of a variance, we cannot meaningfully conclude that apparent improvements achieved by Gorilla (which are often on the order of 1-3% in overall score) are the result of anything other than noise. To be sure, a 7B parameter model that achieves even comparable results to a massive LLM is a notable result (though see my earlier point about fine-tuned baselines), but I would temper claims about improvement in the absence of statistical justification.

Lastly, I feel that the paper would benefit greatly from a more robust error analysis. In the best-performing realistic setting (i.e. without access to an oracle), Gorilla achieves an overall accuracy of roughly 65%. This means that more than one third of the time it is returning an incorrect result (including errors, the authors point out, that could be silent to the end user). Obviously this is far from being ready for direct deployment without external validation. Given the prevalence of errors, it would be useful to better understand what kinds of errors are common: are they more likely to occur with particular kinds of APIs? Does the wording and / or complexity of the input prompt have a large effect? This is the kind of analysis that, in combination with the novel dataset, could help spur future work and improvement. In addition, given the potential for damaging errors, a more robust broader impact statement would be beneficial.

**Questions:**

How do models which are fine-tuned compare to Gorilla? (If such a comparison is not appropriate, why?)

What is the statistical significance of the results?

What are the most common kinds of errors?

Do the kinds of errors differ between the different categories of APIs?

As a final minor question: do the authors have an explanation for why GPT-3.5 appears to outperform GPT-4 across a few different problem settings? This seems like a surprising and somewhat notable result!

**Limitations:**

I feel that the authors should have a more robust discussion of the broader impact of their work.

---

> ### Author Rebuttal · Authors · 2024-08-07
>
> We thank the reviewer for their time and their thoughtful comments. We are encouraged that you found our choice of model and dataset useful to the larger community. We will clarify some of the questions below:
>
> **1. Fine-tuned version of another model and comparison**
>
> We agree that GPT and Claude models are generalist models. We have evaluated the performance of the model across three diverse base models : MPT-7B (0.70), Falcon-7B (0.66) and LLAMA-7B (0.71) on the HF dataset. The results are included in the paper (Appendix Figure 13) and demonstrate that for the same train-eval dataset, our innovate Retrieval Aware Training (RAT) fine-tuning recipe is robust to the underlying base model.
>
> Toolformer, by it’s construction, requires the LLM to reason over the output which extends well for general purpose chat-style queries such as question answering, Wikipedia search engine, etc. Note that Gorilla supports any API call, and doesn’t rely on the output of the API to train the model - which is often not available. Further, Toolformer required a complete re-training (fine-tuning) when the API specification change. However, since Gorilla is trained with RAT, it is robust to changes in API documentation.
>
> The strongest API-focused model when writing the paper is DocPrompting Zhou et al. (ICLR 2023) which looked at choosing the right subset of code including API along with a retriever. We demonstrate that when comparing Gorilla (vs) DocPrompting - which is a fine-tuned model, Gorilla improves accuracy, while lowering the hallucination. We are happy to include any new baselines the reviewer suggests.
> |Accuracy ↑ | Accuracy ↑ | Hallucination ↓ | Hallucination ↓ |
> |---|---|---|---|
> |DocPrompting | **Gorilla** | DocPrompting | **Gorilla**|
> |61.72 |**71.68**| 17.36| **10.95**|
>
> **2. What is the statistical significance of the results?**
>
> Thank you for highlighting this. To demonstrate that Gorilla’s novel Retriever Aware Training (RAT) helps the model generalize very well to out of domain, for the rebuttal we evaluated RAT trained Gorilla on 2000 APIs from a totally diverse set including hyperscalers (GCP, Azure, AWS), Postman, RapidApi etc,  and find that Gorilla (acc: 0.89) outperforms gpt-3.5-turbo-0125 (acc: 0.75). Some examples in this new set are: `stripe.Charge.create(amount={amount}, currency={currency}, source={source}, description={description})` to create a charge in Stripe, and `yfinance.Ticker({ticker}).dividends` to get dividend information of a stock from Yahoo Finance, which are completely out-of-domain to ML APIs demonstrating the ability to generalize well.
>
> **3. Understanding the errors**
>
> Thank you for your suggestion on including an analysis of errors. Given we have all the generations, this is an easy addition to the paper. In the interest of space, we present one example here, which also highlights errors that are possible even with an oracle retriever. TorchHub APIs usually require pre-processing that is often not templated like in HuggingFace. This causes quite a bit of confusion for the model. For example, to load Tacotron 2 from Nvidia, the right API call is tacotron2_model = torch.hub.load('NVIDIA/DeepLearningExamples:torchhub', 'nvidia_tacotron2', model_math='fp16'). However, the model confuses this with the pre-processing step which looks very similar utils = torch.hub.load('NVIDIA/DeepLearningExamples:torchhub', 'nvidia_tts_utils'), returning the pre-processing API instead of the model call.
>
> In terms of categories, common types of errors include model-name hallucination `resnet-201` instead of `resnet-101`, hallucination in API call format from `torch.hub.load` to `torch.load`, chanes in directory structure e.g. `'facebookresearch/pytorch_GAN_zoo:hub` to `facebook/pytorch_GAN_zoo:hub`. The error categories vary in how often they occur as we go from 0-shot to BM-25 to oracle-retriever, but they remain steady as we vary datasets.  We will include a collection of examples for more qualitative analysis in the appendix of the paper which would be a rich playground for future work!
>
> **4. Why GPT-3.5 appears to outperform GPT-4 across a few different problem settings**
>
> This is a great point of interest even for us. One potential explanation we got when we reached out to OpenAI is that GPT-4 is “lightly” RLHF’d when compared to GPT-3.5. Making 3.5 a stronger instruction follower. So, we can observe that as we move from 0-shot to oracle-retriever, 3.5 shines, since it is better at “following the instruction” of using the API to answer the user question, while GPT-4 on the other hand, tends to hallucinate even when provided with an API - oracle or not.

---

> > ### Comment · Reviewer_jdAv · 2024-08-11
> >
> > Thank you for the response. I appreciate the clarification about why Toolformer would not be an appropriate direct comparison. A few questions and concerns remain, however.
> >
> > First, I'm not sure I was totally clear about question (2) -- I am asking for some kind of a t-test to compare the performance of different models. Are the improvements from Gorilla statistically significant (i.e. P < 0.05)? This is not a question of out-of-domain generalization, but of the robustness of the apparent improvement.
> >
> > For question (3) -- while I understand that constraints on space make including additional examples difficult, I would recommend at least a supplemental table in which the counts of different error types (e.g. "model name hallucination" or "formatting hallucination" as you present) are presented.

---

### Official Review · Reviewer_Umrp · 2024-07-11

**Soundness:** 2
**Presentation:** 3
**Contribution:** 1
**Rating:** 6
**Confidence:** 4

**Summary:**

This paper introduces Gorilla, a fine-tuned LLaMA model designed to improve large language models' ability to use APIs accurately. The authors created APIBench, a comprehensive dataset of ML APIs from HuggingFace, TorchHub, and TensorFlow Hub, and used self-instruct to generate instruction-API pairs for training. They fine-tuned LLaMA-7B with retrieval-aware training, incorporating tech like AST sub-tree matching for evaluating API call accuracy, retriever-aware training to adapt to API changes, and handling of constrained API calls. Results show that Gorilla outperforms existing LLMs (including GPT-4) on API call accuracy across multiple datasets, demonstrates ability to adapt to test-time changes in API documentation, and handles constrained API calls effectively. The paper presents a novel approach to improving LLMs' API usage capabilities, with promising results that outperform existing state-of-the-art models in this specific domain.

**Strengths:**

* Gorilla outperforms existing state-of-the-art language models, including GPT-4 and Claude, in API call accuracy across multiple datasets (TorchHub, HuggingFace, and TensorFlow Hub).
* Gorilla significantly reduces API argument hallucination errors compared to other models, improving the reliability of API calls.
* The retriever-aware training enables Gorilla to adapt to test-time changes in API documentation, allowing it to remain up-to-date with evolving APIs without requiring retraining.
* Gorilla demonstrates the ability to understand and respect constraints (e.g., accuracy requirements) when selecting appropriate APIs, outperforming other models in constraint-aware API invocations.

**Weaknesses:**

A significant weakness of the Gorilla approach is that it relies heavily on knowledge augmentation, which is a common technique already used in various domains to improve language model performance on specific tasks. The use of retrieval-augmented generation to enhance API calling capabilities doesn't represent a novel improvement or implementation compared to similar approaches in other domains.

Essentially, Gorilla applies existing knowledge augmentation techniques to the specific task of API invocation, rather than introducing a fundamentally new method for improving language model capabilities. While the results show improvements in API calling accuracy, the core approach of combining retrieval with language model fine-tuning is not innovative in itself. This limits the broader impact and generalizability of the work beyond the specific domain of API invocation.

The lack of significant methodological innovation suggests that similar performance improvements could potentially be achieved by applying existing retrieval-augmented generation techniques to the API calling task, without necessarily requiring the specific Gorilla architecture. This raises questions about the uniqueness and broader applicability of the Gorilla approach beyond the narrow domain explored in the paper.* While Gorilla performs well on the specific API datasets it was trained on, it's unclear how well it would generalize to entirely new APIs or domains not covered in the training data.

Other Notes:
* Although Gorilla shows some ability to handle constraints, its performance in this area is not significantly better than other models like GPT-3.5, suggesting room for improvement.
* The paper doesn't compare Gorilla's performance with specialized API documentation tools or code completion systems, which might be more tailored for this specific task.
* The paper doesn't provide an in-depth analysis of the cases where Gorilla fails, which could provide insights into its limitations and areas for improvement.

**Questions:**

* How does Gorilla's performance compare to other specialized code generation or API-focused models, not just general-purpose LLMs?
* What is the performance impact of varying the size of the training dataset or the base model size (e.g. using LLaMA-13B instead of 7B)?
* How does Gorilla perform on more complex multi-step API workflows rather than just single API calls?

**Limitations:**

NA - Appropriately discussed limitations.

---

> ### Author Rebuttal · Authors · 2024-08-07
>
> We thank the reviewer for their time and thoughtful comments. We are motivated that you found Gorilla can significantly reduce API argument hallucination errors compared to other models, the retriever-aware training to be a novel contribution that allows Gorilla to adapt to test-time changes in API documentation, allowing it to remain up-to-date with evolving APIs without requiring retraining, and that Gorilla demonstrates the ability to understand and respect constraints (e.g., accuracy requirements) when selecting appropriate APIs, outperforming other models in constraint-aware API invocations.
>
> We clarify some of the questions below:
>
> We agree with the reviewer that prior work including Vicuna, Orca [Mukherjee et al.], Textbooks Are All You Need [Gunasekar et al.] have demonstrated knowledge augmentation techniques. However, our study has been centered around connecting LLMs with APIs. We,
>
> 1. Develop and manually curate a high-quality dataset that is valuable to the community.
> 2. Introduce Retriever-Aware Training (RAT) which teaches the LLM to either use or ignore the retrieved context. This is a novel approach that helps Gorilla achieve superior performance.
> 3. Propose evaluation metrics, including the first technique to define and measure hallucination for the domain of (functions and tools) APIs using Abstract Syntax Tree (AST). We also perform human evaluation to validate the competency of the metric.
> 4. Study realistic API calls with constraints (e.g., “I would like to detect pedestrians and cars on the street. I want a model to run on a Raspberry PI with accuracy at least 65% on ImageNet”. Here, Gorilla needs to reason about the constraints - that the model needs to fit the memory budget while still meeting accuracy budgets.)
> 5. Measure hallucination for popular models for APIs.
>
> **1. Generalization to out of domain**
>
> Gorilla’s novel Retriever Aware Training (RAT) helps the model generalize very well to out of domain. For example, for the rebuttal we evaluated RAT trained Gorilla on 2000 APIs from a totally diverse set including hyperscalers (GCP, Azure, AWS), Postman, RapidApi etc,  and find that Gorilla (acc: 0.89) outperforms gpt-3.5-turbo-0125 (acc: 0.75). Some examples in this new set are: `stripe.Charge.create(amount={amount}, currency={currency}, source={source}, description={description})` to create a charge in Stripe, and `yfinance.Ticker({ticker}).dividends` to get dividend information of a stock from Yahoo Finance, which are completely out-of-domain to ML APIs demonstrating the ability to generalize well.
>
> ** 2. Although Gorilla shows some ability to handle constraints, its performance in this area is not significantly better than other models like GPT-3.5, suggesting room for improvement.**
>
> We agree with the reviewer, and highlight that GPT-3.5 is a closed source model about which little is known. With Gorilla, we propose techniques to fine-tune a 7B parameter model that can match if not beat performance of GPT-3.5.
>
> **3. How does Gorilla's performance compare to other specialized code generation or API-focused models, not just general-purpose LLMs?**
>
> The strongest API-focused model when writing the paper is DocPrompting Zhou et al. (ICLR 2023) which looked at choosing the right subset of code including API along with a retriever. We demonstrate that when comparing Gorilla (vs) DocPrompting, Gorilla improves accuracy, while lowering the hallucination. We are happy to include any new baselines the reviewer suggests.
>
> |Accuracy ↑ | Accuracy ↑ | Hallucination ↓ | Hallucination ↓ |
> |---|---|---|---|
> |DocPrompting | **Gorilla** | DocPrompting | **Gorilla**|
> |61.72 |**71.68**| 17.36| **10.95**|
>
> **4. In-depth analysis of failure cases**
>
> Thank you for your suggestion on providing an in-depth analysis of the cases where Gorilla fails, which could provide insights into its limitations and areas for improvement. We will include this in the paper. In the interest of space, we provide an example here:
> TorchHub APIs usually require pre-processing that is often not templated like in HuggingFace. This causes quite a bit of confusion for the model. For example, to load Tacotron 2 from Nvidia, the right API call is `tacotron2_model = torch.hub.load('NVIDIA/DeepLearningExamples:torchhub', 'nvidia_tacotron2', model_math='fp16')`. However, the model confuses this with the pre-processing step which looks very similar `utils = torch.hub.load('NVIDIA/DeepLearningExamples:torchhub', 'nvidia_tts_utils')`, returning the API for pre-processing instead of the model call.
>
> **5. What is the performance impact of varying model**
>
> We have evaluated the performance of the model across three diverse base models : MPT-7B (0.70), Falcon-7B (0.66) and LLAMA-7B (0.71) on the HF dataset. The results are included in the paper (Appendix Figure 13) and demonstrate that all kept constant, our innovative Retrieval Aware Training (RAT) fine-tuning recipe is robust to the underlying base model.
>
> **6. How does Gorilla perform on more complex multi-step API workflows rather than just single API calls?**
>
> Gorilla is focused at improving the LLM model to invoke single round APIs. Multi-step APIs is exciting future work and outside the scope of this paper.

---

> > ### Comment · Reviewer_Umrp · 2024-08-08
> >
> > To keep the authors updated:
> >
> > Thank you for providing an in-depth response to my queries. I appreciate your clear rebuttals to my concerns, and tangible commitments to revise the work based on the feedback.
> >
> > I will read the rebuttal again, and respond with follow-up queries (if applicable). I intend to change my scores based on my improved understanding.

---

> > > ### Comment · Reviewer_Umrp · 2024-08-12
> > >
> > > I have updated the score, based on the rebuttal presented by the author.

---

### Official Review · Reviewer_MqVN · 2024-07-15

**Soundness:** 3
**Presentation:** 3
**Contribution:** 4
**Rating:** 7
**Confidence:** 4

**Summary:**

This paper addresses a pipeline to call adequate APIs among massive pools to accomplish users’ instructions. For that, the authors construct and release the APIBench dataset that contains more than 1645 APIs, and propose the Gorilla model, which is a retrieval-aware finetuned Llama-7B model for API calls.

**Strengths:**

- Constructing the APIBench with AST tree matching evaluation metrics as well as open-sourcing the trained Gorilla model will be largely benefit to the community.
- The problem in this paper is timely in terms of LLM applications and eco-systems — i.e., automatizing API function calls.
- The fine-tuned model, Gorilla, surpasses performance compared to current SOTA models.
- The paper is well written and soundly provides experiment results and analysis.

**Weaknesses:**

- Lack of details about the training data for Gorilla, such as data stats and construction methods. It’s unclear how different the Gorilla training data and evaluation sets of APIBench are, which could lead to doubts about data contamination.
- Experiment results analysis
    - How robust are models to paraphrased user instructions corresponding to the same target model API?
    - Providing performance for each domain could make it possible to analyze which domains are easy and hard.
- Clarifications are needed:
    - In the experimental result section, the authors report two metrics along with overall accuracy: **the error by hallucination and by selecting the wrong API call.** The metric equations or explanations for calculating the values need to be clarified.
    - In section AST as a Hallucination Metric (line 251), how was experimented to attain the human evaluation of 78% accuracy?
- It is required to mention the license of crawled APIs for usage.
- A limitation section is absent, though the authors describe them in the checklist. Broader impact and limitation could include implicit risks such as usage of unreliable APIs, license infringement, or unexpected results by incorrect function calls.

**Questions:**

- In section 4.1, “Note that for TorchHub and TensorHub, we evaluate all the models using AST tree accuracy score. *However, for HuggingFace, since the dataset is not exhaustive, for all the models except Gorilla, we only check if they can provide the correct domain names.*” is vague. The numbers of data for HF, TFH, and TH are 925, 801, and 95, respectively. Please clarify this.
- Table 1 shows that TorchHub is hard to even with oracle API documents compared to HuggingFace and TensorFlowHub. Can you draw the reason?
- (Typos) line 225. `his suggests`
- (suggestions) Please elaborate more on Table 2 — e.g., the caption and highlights. Moreover, it’s hard to connect between the written explanation and the table, due to different numbers.
- (suggestions) Absence of mentioning the full name of AST (Abstract Syntax Tree)

**Limitations:**

- Limitation section is absent, though the authors describe them in the check list. Broader impact and limitation could include implicit risks such as usage of unreliable APIs, license infringement, or unexpected results by incorrect function calls.

---

> ### Author Rebuttal · Authors · 2024-08-07
>
> We thank the reviewer for their time and thoughtful comments. We are encouraged that you find our contributions of APIBench, and AST tree matching evaluation metric, along with open-sourcing of the LLM models to be valuable contribution to the community.
>
> We clarify the questions below:
>
> **1. Details on Gorilla Training: Data and construction and Clarification on Sec 4.1**
>
> The HuggingFace platform hosts and servers about 203,681 models. Post filtering for poor documentation, lack of dependencies, and those that have no information in their model card, etc, we pick the top 20 models from each domain. Post filtering, we arrive at a total of 925 models from HuggingFace. Note that some of the Model APIs are actually a family of APIs. For example, https://huggingface.co/docs/transformers/model_doc/resnet has `TFResNetModel` and `FlaxResNetModel` among others. In those scenarios, we decouple them to be independent APIs. Similarly, TensorFlow Hub is versioned into v1 and v2. The latest version (v2) has 801 models in total. After filtering, we are left with 626 models. Similarly, we extract all 95 models (exhaustive) from Torch Hub.
>
> Post extraction, we manually check every data-point for quality and correctness. For each API, we verify the dataset to ensure it is executable. Since our evaluation metric checks against the ground truth, having a correct answer using our metric is guaranteed to have high quality code. (Please also see our human eval, highlighting the relationship between the evaluation metric with the final execution accuracy.)
>
> Then, guided by the self-instruct paradigm Wang et al. (2022a), we employ gpt-4-0613 to generate synthetic instruction data. We provide three in-context examples, along with reference API documentation, and task the model with generating real-world use cases that call upon the API. We specifically instruct the model to refrain from using any API names or hints when creating instructions. We constructed 6 examples (Instruction-API pairs) for each of the 3 model hubs. These 18 examples were the only hand-generated data. For each of our 1,645 APIs, we generate 10 instruction-API pairs by sampling 3 of 6 corresponding instruction examples in each pair (illustrated in Figure 3). We then divide the data into a train, and test split (uniform) randomly.
> We train Gorilla for 5 epochs with the 2e-5 learning rate with cosine decay, with batch size 64, warm-up ration 0.03, and 2048 max sequence length.
>
> **2. How robust are models to paraphrased user instructions corresponding to the same target model API?**
>
> The models trained can respond to a diverse set of user instructions, given the diversity in our training data-set. As a yard-stick Gorilla models have been downloaded 10000+ times, and here are some examples randomly chosen:
> ```
> Example 1 (q 638) “I am an illustrator, I want to create an appealing image based on a text description for commercial purposes.”
>
> Example 2 (q 704) “Assist me in finding the accurate information in a table related to the NYSE stock market”
>
> Example 3 (q 867)  “We want to communicate product information to online customers in France. 'Introducing the new eco-friendly water bottle made of high-quality stainless steel with double-wall insulation to keep your drinks cool for 24 hours or hot for 12 hours'.”
> ```
>
> **3. Providing performance for each domain could make it possible to analyze which domains are easy and hard.**
>
> This is a very valuable feedback and have already included this in the paper. We find the variance to be impacted by the number of models in the category. For, example, in HuggingFace, `Computer Vision Object Detection` has an accuracy of 0.77 compared to `NLP Text2Text Generation` which has an accuracy of 0.68.
>
> **4. The metric equations or explanations**
>
> Thank you for raising this. We will clarify in the paper. Here’s a brief explanation: We first parse all the API calls in the API pool using Abstract Syntax Tree (AST). We then parse the LLMs output into the AST. We then check whether the LLMs output matches any AST (subtree match) in the pool. Note that we not only check the functional name but also the non-optional arguments. For example, in HuggingFace, one example could be image classification: pipeline('image-classification', model='fxmarty/resnet-tiny-beans'), we check the AST tree node function argument pipeline, value argument image-classification and fxmarty/resnet-tiny-beans. If they all matches the model’s output, we claim the model’s output matches this specific API. We then check the function description of model’s output and the matched API in our pool. If they are equivalent, we claim correct. Evaluation metrics: Accuracy is the #Correct / #Total. In order to be correct, the model’s output not only needs to match one API in the API pool, but the matched API’s function description needs to be the same as the ground truth. Hallucination is the #Not Matched / # Total. This simply means the model’s output doesn't match any API call, thus we cannot find the corresponding API in our dataset.
> Accuracy + Hallucination + Error (syntactic, etc) = 1
>
>
> **5. License of crawled APIs**
>
> The licenses of crawled APIs are permissible under Apache 2.0. TensorFlow is licensed under the Creative Commons Attribution 4.0, and code samples are licensed under the Apache 2.0. Pytorch Hub follows the Linux Foundation Policies. The huggingface hubdocs are all on Apache 2.0.
>
> **6.  TorchHub is hard to even with oracle API documents**
>
> TorchHub APIs usually require pre-processing that is often not templated like in HuggingFace. This causes quite a bit of confusion for the model. For example, to load Tacotron 2 from Nvidia, the right API call is `tacotron2_model = torch.hub.load('NVIDIA/DeepLearningExamples:torchhub', 'nvidia_tacotron2', model_math='fp16')`. However, the model confuses this with the pre-processing step which is similar returning  `utils = torch.hub.load('NVIDIA/DeepLearningExamples:torchhub', 'nvidia_tts_utils')`.

---

> ### Author Response · Authors · 2024-08-07
> **How was the experiment conducted to attain the human evaluation of 78% accuracy?**
>
> **7. human evaluation of 78% accuracy?**
>
> This is from human evaluation by directly executing the code. We manually evaluated the generated results on 100 LLM generations (randomly chosen from our eval set). The accuracy using AST subtree matching is 78%. We observe that this is consistent with human evaluation that revealed a 78% accuracy in calling the right API. All the generations that AST flagged as incorrect, were the same ones that were manually also flagged as incorrect. Additionally, Gorilla also generates supporting code to call the API which includes installing dependencies (e.g., `pip install transformers[sentencepiece]`), environment variables, etc. When we manually attempted to execute these codes, 72% of all codes generated were executed successfully. It's worth noting that the 6% discrepancy are NOT semantic errors, but errors that arose due to factors external to the API in the supporting code - we have included an example to illustrate this further. Considering the significant time and effort required for manual validation of each generation, our data further reinforces our belief in the efficiency of using AST as a robust offline metric.
> Here is a representative example, where we are able to load the correct model API. However, in the supporting code, after we have the output from the API, the `zip()` function tries to combine sentiments and scores together. However, since scores is a `float`, it's not iterable. `zip()` expects both its arguments to be iterable, resulting in an `'float' object is not iterable` error.
>
> ```
> from transformers import pipeline
> def load_model():
>    classifier = pipeline('sentiment-analysis', model='nlptown/bert-base-multilingual-uncased-sentiment')
>    return classifier
>
> def process_data(comments, classifier):
>    response = classifier(comments)
>    sentiments = response[0]['label'].split()
>    scores = response[0]['score']
>    result = [{'sentiment': sentiment, 'score': score} for sentiment, score in zip(sentiments, scores)]
>    return result
>
> comments = "These comments are about our news website."
> # Load the model
> classifier = load_model()
> # Process the data
> response = process_data(comments, classifier)
> print(response)
> ```

---

> > ### Comment · Area_Chair_CCqp · 2024-08-12
> > **Reviewer please respond**
> >
> > Dear reviewer,
> >
> > Thank you for your efforts in reviewing this paper. Now that the authors have provided their response, do you have any further comments?
> >
> > Thank you,
> > AC

---

### Author Rebuttal · Authors · 2024-08-07

We are encouraged by the insightful reviews and appreciate the recognition of the key strengths of our work. The reviewers highlighted:
1. The timely relevance of our problem statement in the realm of Large Language Models (LLMs) and API ecosystems, emphasizing our novel approach in automatizing API function calls **[MqVN, jdAv]**.
2. Our model, Gorilla, has been noted for its exceptional performance, surpassing current state-of-the-art (SOTA) models like GPT-4 and Claude in API call accuracy, and significantly reducing API argument hallucination errors **[Umrp, MqVN]**.
3. The novel use of AST tree matching evaluation metrics for measuring LLM hallucination in calling APIs, and the construction of APIBench, as well as the open-sourcing of Gorilla, are particularly noted for their potential to benefit the community **[MqVN, N9sF]**.
4. The paper’s clear, well-structured presentation, detailed comparative analysis, and comprehensive experimental results were also commended **[N9sF, MqVN]**.
5. We acknowledge the positive feedback on the model’s ability to adapt to evolving APIs and its effectiveness in constraint-aware API invocations **[Umrp]**.

We address individual concerns below, and all suggested revisions have been incorporated into the manuscript.

---

### Decision · Program_Chairs · 2024-09-25

**Decision:**

Accept (poster)

**Comment:**

This work proposes Gorilla, a model that is able to achieve state of the art in API call generation. The authors will release the code and data (APIBench, which has AST matching metrics) to the community. The reviewers generally appreciate the work and impact that the model and dataset will have and comment on the clear presentation of the paper, as well as its actual performance compared to the baselines. They do raise some concerns, such as lack of training details, clarifications on metrics and results analysis, novelty, and evaluation (results, statistical significance). However, the authors responses are clear and seem convincing to the reviewers.